# Learning to Abstract with
# Nonparametric Variational Information Bottleneck

**Melika Behjati**[† 1,2]  **Fabio Fehr**[† 1,2]  **James Henderson** [1]

[1] Idiap Research Institute, Switzerland

[2] École Polytechnique Fédérale de Lausanne, Switzerland

`firstname.lastname@idiap.ch`

## Abstract

Learned representations at the level of characters, sub-words, words and sentences, have each contributed to advances in understanding different NLP tasks and linguistic phenomena. However, learning textual embeddings is costly as they are tokenization specific and require different models to be trained for each level of abstraction. We introduce a novel language representation model which can learn to compress to different levels of abstraction at different layers of the same model. We apply Nonparametric Variational Information Bottleneck (NVIB) to stacked Transformer self-attention layers in the encoder, which encourages an information-theoretic compression of the representations through the model. We find that the layers within the model correspond to increasing levels of abstraction and that their representations are more linguistically informed. Finally, we show that NVIB compression results in a model which is more robust to adversarial perturbations.

## 1 Introduction

Learning representations of language using self-supervision has become a cornerstone of NLP (Pennington et al., 2014; Peters et al., 2018; Devlin et al., 2019, *inter alia*). However, these representations are specific to their tokenisation (e.g. Byte-Pair (Sennrich et al., 2016), WordPiece (Schuster and Nakajima, 2012), SentencePiece (Kudo and Richardson, 2018), characters (Al-Rfou et al., 2019), and even bytes (Xue et al., 2022)), which restricts the level of abstraction from the input text which their representations are able to convey. Work like CANINE (Clark et al., 2022) and Charformer (Tay et al., 2022) avoid problems with tokenisation by modeling individual characters or bytes, and thereafter use a stride-based downsampling to reduce the representation length. The

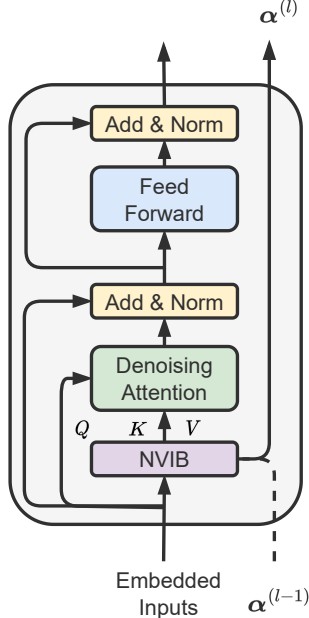

Figure 1: Transformer encoder layer $(l)$ including the NVIB layer and Denoising self-attention module.

stride pattern is fixed and thus can't be considered as learning to abstract. Behjati and Henderson (2023) recently introduced the task of learning a higher level of abstraction in a set-of-vector space by proposing Dynamic Capacity Slot Attention. In this work, we propose a novel character-level model of representation learning which learns different levels of abstraction in different layers of the same model.

**Contributions** We adapt the Nonparametric Variational Information Bottleneck regulariser (NVIB) (Henderson and Fehr, 2023) for application to self-attention in the stacked layers of a Transformer encoder.[1] The resulting model has greater abstraction than a standard Transformer due to selectively dropping some vectors in higher attention layers. Interestingly, we observe that the learned abstract units

---

[†]Equal contribution.

[1]The code is publically available at:
https://github.com/idiap/nvib
https://github.com/idiap/nvib_selfattention

are intuitive, often corresponding to words. By employing different analysis methods, we demonstrate that our model is better at encoding semantically and linguistically meaningful information than a standard Transformer baseline. Moreover, it exhibits an enhanced level of robustness, further consolidating its advantage.

## 2 The Model

Our model consists of standard Transformer encoder-decoder layers (Vaswani et al., 2017), where the encoder block has been augmented with an NVIB regulariser on the self-attention layers, as seen in Figure 1.

### 2.1 NVIB for Self-Attention

Nonparametric Variational Information Bottleneck is an information-theoretic regulariser for attention-based latent representations (Henderson and Fehr, 2023). It has been shown to induce smooth and sparse latent representations in the cross-attention layer of a Transformer encoder-decoder, where Henderson and Fehr (2023) used it to define a Variational Auto-Encoder (VAE) (Kingma and Welling, 2014). It generalises attention over a set of vectors to *denoising attention* over a mixture of impulse distributions, and uses Bayesian nonparametrics to handle the fact that the number of vectors grows with the length of the text. NVIB uses Dirichlet Processes (DPs) to define distributions over these mixture distributions, and controls the information in the latent representation by sampling a mixture distribution from the attention layer's DP, thereby adding noise which removes information.

We extend the previous work by using implicit reparameterisation gradients (Figurnov et al., 2018) to improve learning, and by adapting NVIB for use in the stacked self-attention layers of a Transformer encoder. By extending NVIB's information-theoretic regularisation to the series of latent representations inside the Transformer encoder, we see increasingly abstract interpretable representations in the higher layers.

**NVIB layer**  As with a standard attention layer, an NVIB layer maps a set of $n$ vectors to an attention function. It first maps the $n$ vectors $\boldsymbol{Z} \in \mathbb{R}^{n \times p}$ to the parameters of a DP, which are a total pseudo-count for its Dirichlet distribution and a mixture of Gaussians for its base distribution. Each of the $n$ vectors is individually projected to a pseudo-count $\boldsymbol{\alpha} \in \mathbb{R}^n$ and a Gaussian component

($\boldsymbol{\mu} \in \mathbb{R}^{n \times p}, \boldsymbol{\sigma} \in \mathbb{R}^{n \times p}$) of the base distribution. The model can drop entire vectors by setting their pseudo-counts to zero, thereby making the representation sparse. In addition, there is an $n+1^{th}$ component of the base distribution for the prior, with parameters $\alpha^p=1$, $\boldsymbol{\mu}^p=\boldsymbol{0}$ and $\boldsymbol{\sigma}^p=\boldsymbol{1}$. The individual pseudo-counts are both summed to get the DP's total pseudo-count and normalised to weight the components of the DP's base distribution. The NVIB layer then uses denoising attention to access either a set of weighted vectors sampled from the DP (at training time), or the base distribution of the DP (at testing time).

Henderson and Fehr (2023) use ReLU, linear and exponential activation functions to compute $\boldsymbol{\alpha}$, $\boldsymbol{\mu}$ and $\boldsymbol{\sigma}$, respectively. To adapt NVIB for stacked layers of self-attention, our model replaces the activation for the pseudo-count parameters with an exponential activation, and includes a multiplicative skip connection from the previous layer $l-1$, as shown in Figure 1:

$$\boldsymbol{\alpha}^{(l)} = \exp(\boldsymbol{w}\boldsymbol{Z}^T + b + \log(\boldsymbol{\alpha}^{(l-1)})), \quad (1)$$

where $\boldsymbol{w} \in \mathbb{R}^{1 \times p}$ and $b \in \mathbb{R}$ form the linear projection. The exponential activation allows the model to be more stable in training.[2] The skip connection in between layers $l-1$ and $l$ helps coordinate the importance of vectors across layers. Keeping the pseudo-count parameters in log-space prevents overflow and improves precision when the parameters get larger. This results in a multiplicative skip connection which emphasizes the communication between layers.

To compute self-attention, the DP parameters projected from all the individual vectors together define a single DP, and we take a single sample from this DP which all the individual vectors access via denoising attention. The queries for this denoising self-attention are computed from the original $n$ vectors $\boldsymbol{Z} \in \mathbb{R}^{n \times p}$, before the NVIB layer. We also introduce the use of implicit reparameterisation gradients (Figurnov et al., 2018) for error backpropagation through the sampling step. See Appendix D for the exact attention functions.

**Training objective**  The NVIB loss regularises the attention-based representations so that the size of the representation at each layer is appropriate for the complexity of the representation being encoded at that layer. It has three terms, a reconstruction

---

[2]Since the exponential function is never exactly zero, we threshold small values to introduce sparsity. See Appendix A.

loss $L_R$, and two KL divergence terms: $L_D$ for the pseudo-counts of the Dirichlet distributions, and $L_G$ for the parameters of the Gaussian components. The $L_R$ term is the supervised learning objective, which tries to make the latent representation informative enough to predict the original text. The $L_G$ term tries to make the individual Gaussian components less informative, as in vector-space VAEs (Kingma and Welling, 2014). The $L_D$ term tries to push down the total pseudo-count, which pushes some of the individual pseudo-counts to zero, thereby effectively dropping their vectors and reducing the number of vectors in the latent representation. See Appendix C for loss equations.

To apply NVIB to stacked self-attention layers, we want to allow the lower layers to compute with more vectors while encouraging the upper layers to compress to fewer vectors, thereby encouraging abstraction at the higher layers. We therefore weight the loss terms differently at each layer:

$$\mathcal{L} = L_R + \beta^{(l)}(\lambda_D L_D + \lambda_G L_G) \quad (2)$$

$$\beta^{(l)} = \frac{l}{\sum_{l=0}^{N} l} \quad \text{for } l \in \{1, ..., N\} \quad (3)$$

where $\beta^{(l)}$ controls the degree of NVIB regularisation for layer $l$, linearly increasing it for higher layers. If a vector is dropped in the last self-attention layer (i.e. zero pseudo-count), then we also drop that vector in the cross-attention layer to the decoder, but otherwise there is no NVIB regularisation of the cross-attention.

During preliminary experiments, instead of the above formula for $\beta^{(l)}$ we considered a uniform weight, as well as a doubling weight, per layer. These regularisation weights were either too weak or too strong, respectively. The values we considered for the hyperparameter $\lambda_D$ are given in Appendix B. When we increase this regularisation, the characters are grouped into fewer and fewer vectors until all characters are compressed into a single vector, much like a sentence embedding. If we over-regularise, the representations collapse to the uninformative prior representation.

## 3 Related Work

Modeling language at the level of characters has the advantage of providing an end-to-end framework for the models to operate, without the need for tokenization as a preprocessing step (Xue et al., 2022; Ataman et al., 2020; Choe et al., 2019; Al-Rfou et al., 2019; Kawakami et al., 2017). This

is at the cost of longer sequence lengths and the need for greater model depth to reach the understanding level of subword-based models. While CANINE (Clark et al., 2022) and Charformer (Tay et al., 2022) are some attempts to bypass these shortcomings, they do so by fixed architectural design choices. Our work differs in that it allows the model to learn how to abstract and compress the input without a hard-coded abstraction structure. Our inspiration comes from Behjati and Henderson (2023) who introduced the task of learning a higher level of abstraction and proposed a method based on Slot Attention (Locatello et al., 2020) for this purpose. Our work is also related to HM-RNNs (Chung et al., 2017) as it tends to learn a hierarchy of units within its layers, though it does not make discrete decisions on unit boundaries. Our approach to learning meaningful disentangled abstractions by encouraging the models to learn compressed representations through a bottleneck is shared with VAEs (Kingma and Welling, 2014) and other work in that line (Alemi et al., 2017; Higgins et al., 2017).

## 4 Experiments

Our proposed model's abstractness is analyzed qualitatively through attention visualisations (Section 4.2) and quantitatively through a challenging sub-topic classification task (Section 4.3.1). Each layer is probed to analyse the linguistic information captured (Section 4.3) and finally we examine the models' robustness to adversarial, synthetic noise (Section 4.4). We provide additional details of these experiments in the Appendices F to I.

### 4.1 Experimental Setup

**Data** We train all models on the Wikitext-2 (Merity et al., 2017) encyclopedia dataset at the character level, with a noisy character deletion reconstruction objective (Lewis et al., 2020).

**Models** We compare the self-attention representations from a standard Transformer encoder layer and our Transformer encoder layer with NVIB regularisation. We consider models consisting of six stacked Transformer encoder layers to be in line with the base model from Vaswani et al. (2017). For the Transformer decoder we use only 2 layers so that the decoder is not able to compensate for poor embeddings from the encoder. For simplicity of implementation and interpretation, we use only a single attention head. For the NVIB models,

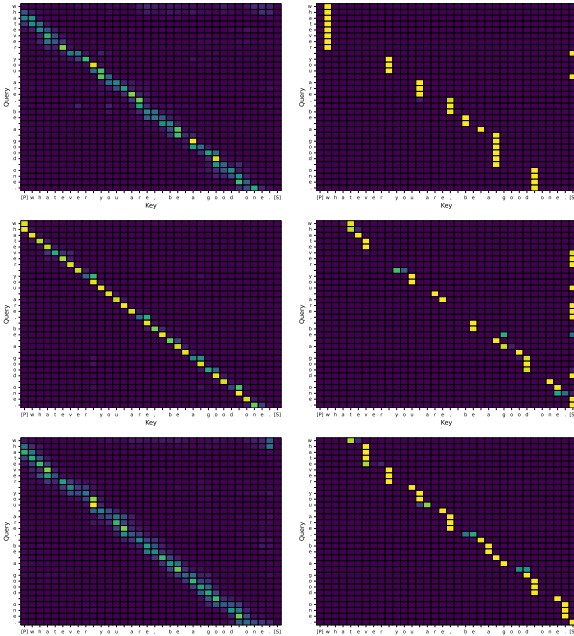

Figure 2: Self-attention patterns of the last 3 layers of 6-layer Transformer encoders from bottom to top. **Left**: Standard self-attention. **Right**: With NVIB regularisation. **Sentence**: "Whatever you are, be a good one." Dark purple is 0 and light yellow is 1 for attention.

we only apply NVIB to the final three layers. To ensure comparability between our model and the baseline, we train the baseline to have the same denoising capability and thus the same validation cross-entropy when evaluated on noised examples. For further details see Appendices A and B.

## 4.2 Attention Map Visualisations and Analysis

To qualitatively evaluate the model's ability to learn interpretable abstractions, we visualise the self-attention maps. Figure 2 compares the self-attention patterns of the the last 3 layers of: a Transformer with 6 layers of standard attention (left); and a Transformer with 3 layers of standard attention followed by 3 layer of denoising attention with NVIB (right).

Despite being trained solely on noisy reconstruction at the character level, the NVIB layers compress the self-attention representations through the layers into distinct groups. At lower levels, the model uses nearly all vectors (i.e. $\sim 99\%$) and learns position-local information, shown as a diagonal pattern. At higher levels the model drops some vectors (the blank columns) and groups characters (the vertical bars) in ways which strongly resemble subword units or even words. The last

| | P | R | F1 |
|---|---|---|---|
| Transformer | **95.51** | 56.51 | 64.52 |
| NVIB | 85.23 | **79.02** | **78.86** |

Table 1: Word segmentation performance [%].

level retains only an average of $\sim 35\%$ of vectors. This is because the stronger NVIB regularisation at higher layers encourages the grouping of correlated characters, to reduce redundant information, and the strongest correlations are within words. We provide further examples in Appendix E.

We quantify the resemblance of the final-layer self-attention maps to words by extracting contiguous segments from the maps and computing the F1 measure between our segments and the words in the sequence. In particular, we find the best alignment between words and segments and compute the number of characters in the longest common substring between a word and its corresponding discovered segment.[3] Table 1 compares the performance of our model to the Transformer baseline. This impressive unsupervised performance (F1 of $78.86\%$) concurs with the attention visualisations and quantitatively verifies that our model has learned to abstract to the level of words.

## 4.3 Probing Analysis

This section uses different probing tasks to quantitatively evaluate the abstraction capabilities of our model and analyse the linguistic information captured by the layers.

### 4.3.1 ArXiv Topic Classification

The ArXiv topic classification task (Hofmann et al., 2022) is a challenging task consisting of short input sentences with long technical words. For each subject, the classifier should classify the topic into 20 possible sub-areas. Following Behjati and Henderson (2023), we train an attention-based probe on the final layer of the models and report the F1 measure for performance on the ArXiv-L dataset. Without finetuning the models, this classification task serves as probing high-level abstract linguistic properties (Hewitt et al., 2021). As shown in Table 2, the NVIB layer results in the model learning more information about the meaning and semantics in the abstract representations than characters and therefore provides better units for performing the task.

---

[3]See Appendix I for further details and exact formulas.

| Task | Transformer | NVIB |
|---|---|---|
| Computer science | 42.33 | 44.47 |
| Mathematics | 44.02 | 47.13 |
| Physics | 48.83 | 52.32 |
| **Average** | 45.06 | **47.97** |

Table 2: F1 score [%] on Arxiv-L classification task.

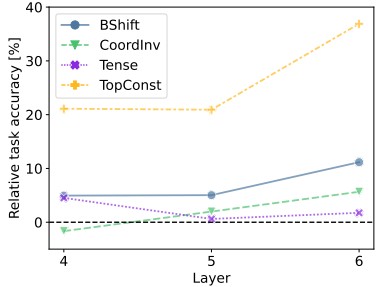

Figure 3: Relative performance of NVIB over Transformer for a subset of SentEval tasks.

### 4.3.2 Linguistic Probing

The SentEval task set is specifically designed to examine the linguistic information available in a sentence representation at different levels, ranging from surface-level to semantic-level tasks (Conneau et al., 2018; Conneau and Kiela, 2018). We probe for linguistic information of our model and the baseline Transformer, across all layers. In general, the performance improves in deeper layers and increases further with the inclusion of NVIB in the layers. We highlight the results of four tasks in Figure 3, which to perform well in these tasks the representations must capture latent syntactic structures (**BShift**), cluster them by constituent types (**TopConst**), or have an understanding of semantics (**Tense**) or broad discourse and pragmatic factors (**CoordInv**) (Conneau et al., 2018). The inclusion of our NVIB layers increases the relative performance over the Transformer baseline, showing it to be more linguistically informed. The complete set of results is in Appendix Table 4.

### 4.4 Robustness Analysis

We analyze the robustness of our models to synthetic noise injected into the input sequences (Belinkov and Bisk, 2017; Durrani et al., 2019). Namely, we evaluate the reconstruction quality when the inputs are perturbed by swapping, deleting, inserting, and substituting characters (Morris et al., 2020). We expect our model to be more robust due to its compressed representations. Figure 4 shows that our model is more robust to adversarial

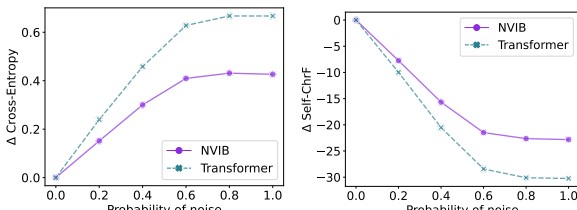

Figure 4: Robustness plots showing relative performance change over increasing input perturbations.

noise than a standard Transformer, with increased advantage as the level of noise increases.

## 5 Conclusions

We propose a novel method for inducing abstract representations of text. We adapt the Nonparametric Variational Information Bottleneck (Henderson and Fehr, 2023) regulariser for application to self-attention in the stacked layers of a Transformer encoder. Our model learns how many vectors are needed at each layer, thereby inducing different levels of abstraction in different layers of the same model. We find that these abstract units are intuitive, more robust, and better at encoding semantically and linguistically meaningful information.

## Limitations

While the models and training data are reasonable in size, the experiments do not include the very large scale training often found in work on representation learning in text. We anticipate that the advantages of NVIB on self-attention layers will only increase as the models and data are scaled up, since this should allow even more abstract representations to be learned. In addition, the experiments are only done on English, but we would expect more improvements with more morphologically rich languages. In future work we plan to explore fine-tuning NVIB for sparsity and downstream performance, and consider different tokenizations beyond characters only.

## Ethics Statement

We foresee no ethical concerns with our work.

## Acknowledgements

Both Melika Behjati and Fabio Fehr were supported by the Swiss National Centre of Competence in Research (NCCR) under the project Evolving Language, grant number "51NF40_180888".

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

# A   Training Details

**General Training**   All models are trained, without pretraining, using the same encoder and decoder configuration for comparability. Our encoder size is defined by the base Transformer (Vaswani et al., 2017) such that we have a six layer Transformer encoder. However, we use a two layer decoder to ensure the task is not learned in the decoder alone. We use a single attention head. The size for the word embedding vectors and model projections are $512$ and feed forward dimensions $512$, which leads to models of approximately 12-17 million trainable parameters. An English character level tokeniser is used for tokenisation with a vocabulary of approximately 100 characters. During training we use: a learning rate of $1e^{-3}$ with a cosine cool-down over all steps, RAdam optimiser (Liu et al., 2020) with mixed precision (FP16), a batch size of $512$, gradient norm clipping $0.1$ and trained for $55$ epochs ($\approx 8K$ steps). The number of steps were selected considering model convergence and minimising computation time. We use a dropout rate of $0.1$. The input is noised at each

batch with a probability of character deletion of 0.1. Each model takes approximately 2.5hrs on a single NVIDIA GeForce RTX 3090.

**NVIB Training** Training the models with the NVIB layers requires regularising the representations. The introduction of the exponential activation function (as opposed to ReLU) for the psuedo-count parameter $\boldsymbol{\alpha}$ requires a threshold at test time to be exactly zero. We use a threshold for this at $0.1$. During training and testing we enforce a bottleneck between the encoder and decoder by masking the final encoder representations by the aforementioned threshold.

The NVIB hyperparameters $\lambda_G$, $\lambda_D$ and $\alpha^\Delta$ are selected through hyperparameter tuning. However, during training we only sample once from each component thus the approximation parameter is set to $\kappa^\Delta = 1$. We use a Kullback-Leibler annealing divergence strategy where the introduction of the KL divergence loss is linearly introduced between $30\% - 60\%$ of the training steps. This allows the model to learn initial representations, slowly introduce the regularisation and finally learn through the compressed latent representation.

## B  Hyperparameter Tuning

The models are trained on the Wikitext-2 training dataset using the loss from Equation 2. They are tuned on the validation dataset with the aim to be able to reconstruct the character sequence from a noisy input. Following from (Henderson and Fehr, 2023) we set the weights of the Gaussian and Dirichlet KL divergences to be independent of the sentence length $n$ and dimensionality of vectors $d$:

$$\lambda_D = \frac{1}{n}\lambda'_D \;\; ; \quad \lambda_G = \frac{1}{d}\frac{1}{n}\lambda'_G$$

where $\lambda'_D$ and $\lambda'_G$ are fixed hyperparameters. All combinations of the following hyperparameters were considered in a grid search for the respective models:

- $lr = \{1e^{-4}, 1e^{-3}\}$

- $\lambda'_G = \{1e^{-5},\ 1e^{-4},\ 1e^{-3},\ 1e^{-2}\}$

- $\lambda'_D = \{1e^{-2},\ 1e^{-1},\ 1\}$

- $\alpha^\Delta \;\; = \{0\,, 0.05,\ ...,\ 0.45,\ 0.5\}$

where $\lambda'_G$ and $\lambda'_D$ are the weights on the Gaussian and Dirichlet KL divergences. The $\alpha^\Delta$ represents the conditional prior parameter. The final models'

hyperparameters are reported in Table 3 where the validation cross-entropy (CE) is matched for NVIB and baseline Transformers.

| | Transformer | NVIB |
|---|---|---|
| NVIB layers | - | 3 |
| $\lambda_G$ | - | $1e^{-2}$ |
| $\lambda_D$ | - | 1 |
| $\alpha^\Delta$ | - | 0.25 |
| Training Steps | $2.5K$ | $8K$ |
| Val. CE | 0.19 | 0.19 |

Table 3: Hyperparameters for final models evaluated.

The encoders 6 layers are inspired by the base model of Vaswani et al. (2017). For the Transformer decoder we use only 2 layers such that the decoder is not able to overpower the embeddings from the encoder it sees through cross attention.

**NVIB Configuration** For the NVIB layers during experimentation we considered: All layer including NVIB; the last 3 layers including NVIB; and only the final layer including NVIB. When all layers were included it was challenging to get both compression and performance as the regularisation was too strong. Only regularising the last layer managed to reduce the number of vectors but often converged to a single sentence vector with lower, non-comparable validation cross-entropy. Finally, we settled on only regularising the last 3 layers as it gave the model enough flexibility in the lower layers and progressive compression in the higher layers.

## C  KL Divergence Loss

Henderson and Fehr (2023) define the Kullback-Leibler divergence for NVIB with two terms: the $L_D$ for the Dirichlet distribution weights defined by $\boldsymbol{\alpha}$; and the $L_G$ for the distribution of vectors $\boldsymbol{Z}$ generated by the Gaussian components. We set the approximation parameter $\kappa_0 = 1$. This gives us the following loss terms for the KL divergence, where $\Gamma$ is the gamma function and $\psi$ is the digamma function:

$$\begin{aligned}
L_D = &\log\Gamma(\alpha_0^q) - \log\Gamma(\alpha_0^{p'}) \\
&+ (\alpha_0^q - \alpha_0^{p'})(-\psi(\alpha_0^q) + \psi(\alpha_0^q)) \quad (4)\\
&+ \left(\log\Gamma(\alpha_0^{p'}) - \log\Gamma(\alpha_0^q)\right)
\end{aligned}$$

where, $\alpha_0^q$ is the sum of all $\boldsymbol{\alpha}$ parameters generated by the NVIB layer. The conditional prior $\alpha_0^{p'} = \alpha_0^p + n\alpha^\Delta$ is controlled by $\alpha_0^p = 1$ and extra pseudo-counts defined by the length $n$ and a hyperparameter $\alpha^\Delta$. The KL divergence between two Gaussians (with diagonal covariance with values $\boldsymbol{\sigma}$ and weighted by the $\boldsymbol{\alpha}$ parameters) is:

$$L_G = \frac{1}{2} \sum_{i=1}^{n+1} \frac{\alpha_i^q}{\alpha_0^q} \sum_{h=1}^{d} \left( \frac{(\mu_{ih}^q - \mu_h^p)^2}{(\sigma_h^p)^2} - 1 \right.$$
$$\left. + \frac{(\sigma_{ih}^q)^2}{(\sigma_h^p)^2} - \log \frac{(\sigma_{ih}^q)^2}{(\sigma_h^p)^2} \right) \quad (5)$$

## D    Denoising attention function

Henderson and Fehr (2023) generalise the set of vectors input to an attention function to a probability distribution over vectors, and generalise attention to a function of these probability distributions called denoising attention.

**Training function**  During training, the set of sampled vectors $\boldsymbol{Z} \in \mathbb{R}^{n \times p}$ and their sampled log-probability weights $\log(\boldsymbol{\pi}) \in \mathbb{R}^{1 \times n}$ are both output by the NVIB layer, thereby specifying the sampled mixture distribution $F$. A set of query vectors $\boldsymbol{u'} \in \mathbb{R}^{m \times p}$ is projected into key space by the grouped matrices $\boldsymbol{W}^Q, \boldsymbol{W}^K \in \mathbb{R}^{p \times d}$ to $\boldsymbol{u} = (\boldsymbol{u'}\boldsymbol{W}^Q(\boldsymbol{W}^K)^T)$. The keys' dimensionality $d$ is used for scaling. Denoising attention can then be computed as:

$$\text{softmax}\left( \frac{1}{\sqrt{d}} \boldsymbol{u}\boldsymbol{Z}^T + \log(\boldsymbol{\pi}) - \frac{1}{2\sqrt{d}}\|\boldsymbol{Z}\|^2 \right)\boldsymbol{Z} \quad (6)$$

For self-attention, we define the original query vectors $\boldsymbol{u'}$ to be the set of vectors input to the NVIB layer, before projecting to DP parameters and sampling.

**Testing function**  During test time, we do not sample $F$, but instead use the mean of the posterior distribution. The test-time denoising attention can then be computed as:

$$\text{softmax}\left( \boldsymbol{u}\left( \frac{\boldsymbol{\mu}}{(\boldsymbol{\sigma}^r)^2} \right)^T + \log(\frac{\boldsymbol{\alpha}}{\alpha_0}) \right.$$
$$- \left( \frac{1}{2}\left\| \frac{\boldsymbol{\mu}}{\boldsymbol{\sigma}^r} \right\|^2 \right)^T - \mathbf{1}_p \left( \log(\boldsymbol{\sigma}^r) \right)^T \right) \quad (7)$$
$$\times \left( \frac{(\boldsymbol{\sigma})^2}{(\boldsymbol{\sigma}^r)^2} \odot (\mathbf{1}_n^T \boldsymbol{u}) + \frac{\sqrt{d}}{(\boldsymbol{\sigma}^r)^2} \odot \boldsymbol{\mu} \right)$$

where $\mathbf{1}_p$ is a row vector of $p$ ones and $(\boldsymbol{\sigma}^r)^2 = (\sqrt{d} + (\boldsymbol{\sigma}^q)^2)$.

## E    Visualisations

In Figures 5 to 8 we include additional visualisations of the self-attention weights.

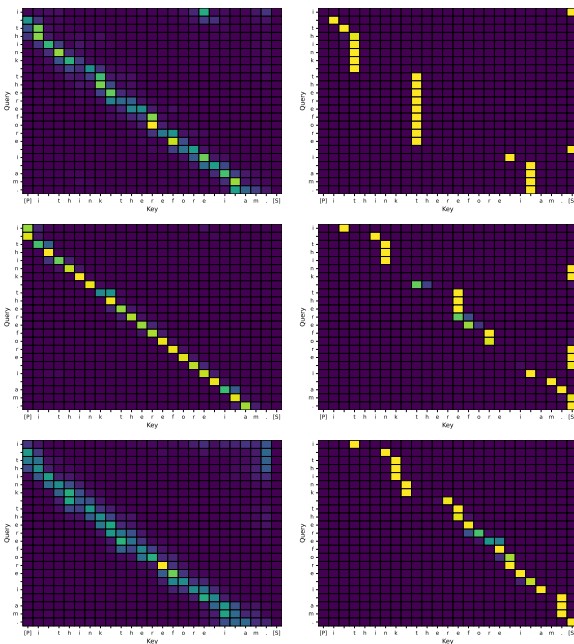

Figure 5: Self-attention patterns of the last 3 layers of 6-layer Transformer encoders. **Left**: Standard self-attention. **Right**: With NVIB regularisation. **Sentence**: "I think therefore I am." Dark purple is 0 and light yellow is 1 for the attention values.

## F    Probing Classifiers

We used two types of probing classifiers to perform our tasks. First, we employ an attention-based probing classifier to operate on top of the set of representations for predicting the specified property. This would be similar to having a learnable [CLS] token which is used as the representation of the sequence in BERT-like models. This is also in line with the findings of Pimentel et al. (2022) that the way we do the probing should resemble how the model itself would use the information within its architecture. In particular, we first map the representations into a new space with a 2 layer MLP. Then, we compute the attention with a learnable query vector. Finally, we linearly project the resulting vector into the number of classes for each task. We refer to this probe as *Attention-based probe*. Second, we tried a less complicated and more common type of probing in which we first aggregate

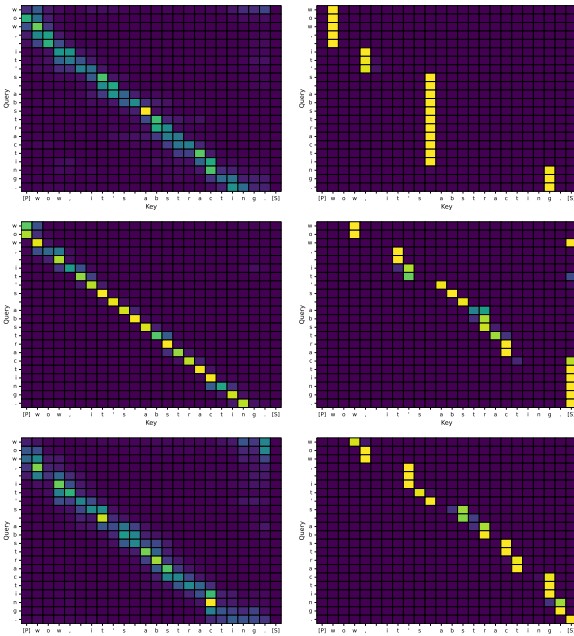

Figure 6: Self-attention patterns of the last 3 layers of 6-layer Transformer encoders. **Left**: Standard self-attention. **Right**: With NVIB regularisation. **Sentence**: "Wow, it's abstracting." Dark purple is 0 and light yellow is 1 for the attention values.

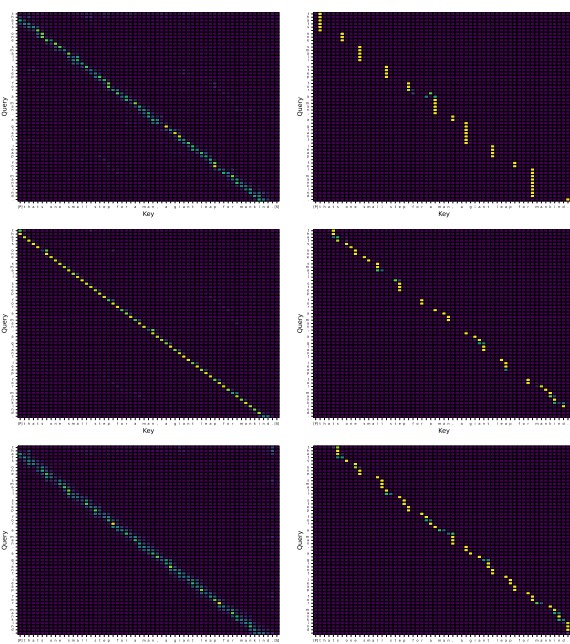

Figure 7: Self-attention patterns of the last 3 layers of 6-layer Transformer encoders. **Left**: Standard self-attention. **Right**: With NVIB regularisation. **Sentence**: "Thats one small step for a man, a giant leap for mankind." Dark purple is 0 and light yellow is 1 for the attention values.

the set of representation vectors by mean and then apply a 2 layer MLP with ReLU non-linearity to

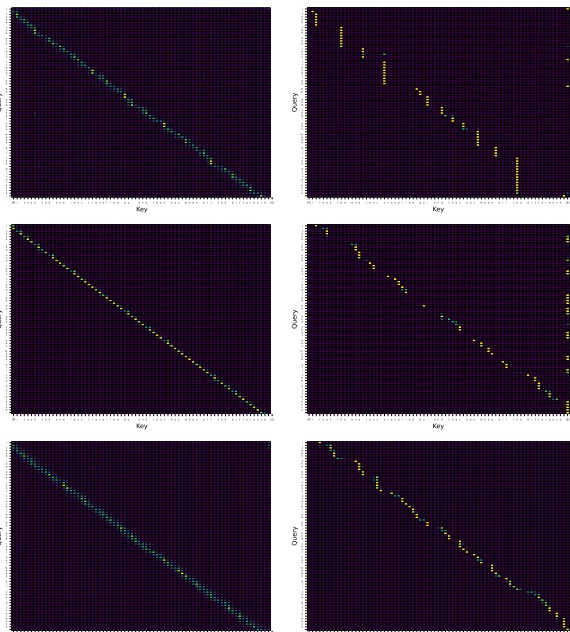

Figure 8: Self-attention patterns of the last 3 layers of 6-layer Transformer encoders. **Left**: Standard self-attention. **Right**: With NVIB regularisation. **Sentence**: "I took the one less travelled by, and that made all the difference." Dark purple is 0 and light yellow is 1 for the attention values.

perform the task. We refer to this probe as *Aggregating probe*.

## G SentEval Tasks

### G.1 Model

We employ the Aggregating probe for performing this task. We froze our models and trained the probes for 10 epochs with a batch-size of 128. The hidden dimension for the probe is set to 256. We trained the model with Adam optimizer with a learning rate of $1e - 4$. We report the test set accuracy for the best-performing model in terms of validation accuracy.

### G.2 Supplementary Results

We report the results in Table 4 on a subset of 7 of the 10 SentEval tasks as sentence length (**SentLen**), word content (**WC**) and semantic odd man out (**SOMO**) tasks are too challenging for our models when encoding from a character level.

## H Arxiv Classification Task

Our goal here is to compare the representations and not have ultimate performance in the task, thus we do not fine-tune the models. Hence, we only

| | Layer | CoordInv | ObjNum | TreeDepth | TopConst | BShift | Tense | SubjNum |
|---|---|---|---|---|---|---|---|---|
| Chance | | 0.5 | 0.5 | 0.125 | 0.05 | 0.5 | 0.5 | 0.5 |
| Transformer | 1 | 0.5023 | 0.6498 | 0.2200 | 0.2880 | 0.5006 | 0.7306 | 0.6500 |
| | 2 | 0.5144 | 0.7255 | 0.2350 | 0.3724 | 0.4994 | 0.7891 | 0.7131 |
| | 3 | 0.5190 | 0.7547 | 0.2594 | 0.4261 | 0.5055 | 0.8263 | 0.7297 |
| | 4 | 0.5196 | 0.7687 | 0.2692 | 0.4368 | 0.5108 | 0.8114 | 0.7545 |
| | 5 | 0.5196 | 0.7737 | 0.2736 | 0.4369 | 0.5304 | 0.8320 | 0.7435 |
| | 6 | 0.5227 | 0.7756 | 0.2736 | 0.4212 | 0.5465 | 0.8384 | 0.7683 |
| NVIB | 1 | 0.5037 | 0.7646 | 0.2349 | 0.3323 | 0.5007 | 0.8344 | 0.7285 |
| | 2 | 0.5069 | 0.7859 | 0.2511 | 0.4243 | 0.5108 | 0.8379 | 0.7777 |
| | 3 | 0.5110 | 0.7963 | 0.2589 | 0.4453 | 0.5466 | 0.8606 | 0.7844 |
| | 4 | 0.5111 | 0.7879 | 0.2655 | 0.5290 | 0.5361 | 0.8481 | 0.7943 |
| | 5 | 0.5299 | 0.7660 | 0.2651 | 0.5283 | 0.5571 | 0.8371 | 0.7793 |
| | 6 | **0.5523** | **0.8207** | **0.2923** | **0.5766** | **0.6075** | **0.8531** | **0.8038** |

Table 4: Performance on Senteval tasks.

evaluated our models on the large division of the task, i.e., ArXiv-L which consist of 1000 samples for each sub-area leading to 20000 samples in total. We employ the Attention-based probe to perform this task as it is quite a challenging task which requires the information in the vectors to be better managed by the Attention mechanism and also more similar to the way the model itself would perform the task. The hidden dimension of the MLP is set to 256 and the query, key, and value matrices are set to the same dimension as the model dimension, namely, 512. We train the classifier with a batch size of 256 for 50 epochs with Adam optimizer with a learning rate of $1e - 3$. Following Hofmann et al. (2022) we report the test F1 for the best-performing model in terms of validation F1.

# I    Quantification of Word Resemblance

We observe a strong resemblance between words and the vertical bands in the final-layer Attention maps of the NVIB integrated model. Therefore, we quantify this similarity as follows. First, we take the $argmax$ over the Key dimension of an Attention map and extract the contiguous segments from the resulting vector. Then, we compute the intersection between the set of obtained segments and the set of words in the sequence. In particular, for each segment and word, we compute the number of intersecting characters (i.e., the length of the longest common substring) as a measure of their overlap. This would lead to a rectangular matrix of scores. Then, we perform the Hungarian matching algorithm (Kuhn, 1955) to find the best 1-1 match between the two sets. Afterward, for each matched word and segment, we compute Precision

$(P)$, Recall $(R)$, and $F1$ measure as

$$P = \frac{\text{longest common substring length}}{\text{segment length}} \quad (8)$$

and

$$R = \frac{\text{longest common substring length}}{\text{word length}}. \quad (9)$$

We reported the average macro $F1$, $P$, $R$ over the validation set of our training data. For the baseline Transformer, as it usually predicts units of length one or two which are within a single word the $P$ would be high as opposed to its $R$ value.