# OpenReview forum: "Learning to Abstract with Nonparametric Variational Information Bottleneck"
_EMNLP/2023/Conference — EMNLP 2023 Findings_

### Official Review · Reviewer_bGq2 · 2023-08-05

**Soundness:** 3

**Excitement:**

3: Ambivalent: It has merits (e.g., it reports state-of-the-art results, the idea is nice), but there are key weaknesses (e.g., it describes incremental work), and it can significantly benefit from another round of revision. However, I won't object to accepting it if my co-reviewers champion it.

**Paper Topic And Main Contributions:**

This paper presents a novel language representation model that learns to compress information to different levels of linguistic abstraction within a single architecture. By applying the NVIB technique, the model achieves a hierarchy of abstraction in its layers and enhances robustness to adversarial perturbations. The contributions lie in the development of this innovative model, its application of NVIB, insights into the hierarchy of abstraction, and its potential to address challenges in linguistic representation learning and adversarial attacks.

**Reasons To Accept:**

1. The paper builds upon prior research by seamlessly integrating Nonparametric Variational Information Bottleneck (NVIB) into the self-attention layers of a Transformer encoder. This adaptation broadens the scope of NVIB's utility in natural language processing tasks.
2. The authors utilize implicit reparameterisation gradients, resulting in more efficient and effective learning processes.
3. The experiments in the paper are comprehensive, and the analyses are thorough.

**Reasons To Reject:**

In my view, reading this article requires prior knowledge of NVIB, which the paper does not provide a brief introduction to. This increases the difficulty for readers who are not familiar with NVIB, but it might not be an issue for experts in the field.

**Reproducibility:**

3: Could reproduce the results with some difficulty. The settings of parameters are underspecified or subjectively determined; the training/evaluation data are not widely available.

**Reviewer Confidence:**

1: Not my area, or paper was hard for me to understand. My evaluation is just an educated guess.

---

> ### Author Rebuttal · Authors · 2023-08-28
>
> We would like to thank Reviewer bGq2 for their time. We believe that Reviewer bGq2 has understood our claims, despite their confidence, and highlighted multiple valid reasons to accept: **innovative**, **comprehensive** and that it **broadens the scope of NVIB’s utility in NLP**. We will address the reviewer's concerns below.
>
> >*The paper does not provide a brief introduction to NVIB*
>
> Due to the limit of 4 pages we needed to be brief regarding the description of NVIB [(Henderson and Fehr, 2023)](https://openreview.net/pdf?id=6QkjC_cs03X).  We think that we conveyed the key properties of our model in Section 2 which are needed to understand the contributions of this paper, and we included more details in Appendix C and D.  However, there are some details which could provide a deeper understanding of why the model works, and if accepted we would be happy to include them given the increased page limit.

---

### Official Review · Reviewer_fPkp · 2023-08-08

**Soundness:** 3

**Excitement:**

2: Mediocre: This paper makes marginal contributions (vs non-contemporaneous work), so I would rather not see it in the conference.

**Paper Topic And Main Contributions:**

This work applies the Nonparametric Variational Information Bottleneck (NVIB) regularizer to the self-attention layer of a transformer model to compress the representations. They demonstrate through various probes and visualizations that the resulting layers learn increasing levels of abstraction for the text. The  authors demonstrate better robustness to noise compared to a base transformer but only for text reconstruction.

**Questions For The Authors:**

1. Have you compared this architecture against the transformer using supervised learning for token/sentence classification or text generation?

**Reasons To Accept:**

1. Interesting idea to use the NVIB regularization to learn "better representations" in a transformer model. The representations can be interpreted to learn various levels of abstraction.


**Reasons To Reject:**

1. The work does not demonstrate any practical benefits of using their method. The analysis shows interesting trends in the representations, but without any experiments on downstream tasks, using either fine-tuning or prompting.  It is hard to imagine if the method is useful. Even if the authors could not pre-train using their method, they could have applied this architecture to a supervised learning problem.

2. A lot of the important paper details are in the appendix. The paper is not sufficiently self-contained.

**Reproducibility:**

3: Could reproduce the results with some difficulty. The settings of parameters are underspecified or subjectively determined; the training/evaluation data are not widely available.

**Reviewer Confidence:**

3: Pretty sure, but there's a chance I missed something. Although I have a good feel for this area in general, I did not carefully check the paper's details, e.g., the math, experimental design, or novelty.

---

> ### Author Rebuttal · Authors · 2023-08-28
>
> We appreciate that Reviewer fPkp finds our proposals **interesting** and that our paper contributes to the topic of interpretability. We address the reviewer's concerns below.
>
> > *The work does not demonstrate any practical benefits of using their method.*
>
> Not every paper at EMNLP needs to be an engineering contribution, especially not a short paper (see [ACL Reviewing Guidelines](https://2023.aclweb.org/blog/review-acl23/#1-acl-2023-review-form), *The authors should have done [X] instead*).  We do not claim practical benefits, which is one reason we submitted to the Interpretability and Analysis of NLP models track.  However, we do demonstrate several properties which will naturally lead to practical benefits in future work, such as sparsity and robustness.  In particular, this includes supervised learning of a downstream task (ArXiv topic classification), just without fine-tuning.
>
> > *Experiments on downstream tasks, using either fine-tuning or prompting.*
>
> **Additional experiments** To address the reviewer's concern, we ran experiments on the ArXiv topic classification task by fine-tuning our pretrained models. Our results are consistent with the ones reported in (Table 1, pg4) in the paper in that our NVIB-integrated model works better than a standard Transformer. We report the F1 performance on ArXiv-L dataset in the Table below. We have an absolute performance gain of 4.03% over the standard Transformer.
>
> | Task      | Transformer | NVIB     |
> | :---        |    :----:   |          ---: |
> | Computer science      | 39.82       | 44.12   |
> | Mathematics   | 43.93        | 47.58      |
> |Physics| 49.18| 53.32|
> | | | |
> |**Average**| 44.31|**48.34**|
>
> > *The paper is not sufficiently self-contained.*
>
> Due to the limit of 4 pages we needed to be brief regarding the description of NVIB [(Henderson and Fehr, 2023)](https://openreview.net/pdf?id=6QkjC_cs03X).  We think that we conveyed the key properties of our model in Section 2 which are needed to understand the contributions of this paper, and we included more details in Appendix C and D for those who are interested.
> However, we would be happy to address the reviewer's concern by moving the most relevant details from the appendix to the main text, given the additional page limit upon acceptance. If the reviewer could be more specific about which sections need additional details then that would help us improve the presentation.

---

### Official Review · Reviewer_2hkB · 2023-08-10

**Soundness:** 3

**Excitement:**

3: Ambivalent: It has merits (e.g., it reports state-of-the-art results, the idea is nice), but there are key weaknesses (e.g., it describes incremental work), and it can significantly benefit from another round of revision. However, I won't object to accepting it if my co-reviewers champion it.

**Paper Topic And Main Contributions:**

The paper introduces a novel method to induce abstract representations of text by adapting the Nonparametric Variational Information Bottleneck (NVIB) regularizer for application to self-attention in the stacked layers of a Transformer encoder. The primary contribution is that the model learns how many vectors are needed at each layer, inducing different levels of abstraction in different layers of the same model. This is achieved without a hard-coded abstraction structure. The paper conducts various experiments, including attention map visualizations, probing analysis, and robustness analysis, to validate the model's capabilities.

**Questions For The Authors:**

The presentation should be improved to make it easier to follow. For example, the right part in Figure 1 is not easy to understand. Does it mean different attention patterns at three different layers? In Figure 2, which color is “lighter” is a challenging task to tell.

**Reasons To Accept:**

[+] Research topic. The research topic of learning different levels of abstraction in the same model is interesting.
[+] Experiments and evaluation. The authors have conducted experiments both qualitatively and quantitatively for performance evaluation.
[+] Presentation. The paper is easy to follow in general. The detailed introduction of the experiment setup is also introduced.

**Reasons To Reject:**

[-] Method: Why applying NVIB to Transformer is challenging is not well-discussed. This is important to demonstrate the technical contribution of this paper.
[-] Experiment: While the paper mentions applying NVIB to the final three layers of the Transformer, it might benefit from a more detailed explanation or exploration of why only these layers were chosen and how different configurations might impact the results.
[-] Baseline: The method is only compared with Transformer, while there are more advanced models for solving the same task. It would be better if some other models can also be compared.


**Reproducibility:**

4: Could mostly reproduce the results, but there may be some variation because of sample variance or minor variations in their interpretation of the protocol or method.

**Reviewer Confidence:**

2: Willing to defend my evaluation, but it is fairly likely that I missed some details, didn't understand some central points, or can't be sure about the novelty of the work.

---

> ### Author Rebuttal · Authors · 2023-08-28
>
> We really appreciate the constructive comments from Reviewer 2hkB, and for taking the time to understand our contributions. We would like to thank the reviewer for the kind remarks by highlighting our **interesting** research topic,  our **qualitative and quantitative experiments**, and **easy-to-follow** presentation.
>
> We believe the constructive concerns of the Reviewer can be addressed and included in future versions of the paper (or Appendix).
>
> >*Method: why applying NVIB to Transformer is challenging?*
>
> We believe that one of our contributions is to take the complex theory (and code) from [Henderson and Fehr (2023)](https://openreview.net/pdf?id=6QkjC_cs03X) and frame its application to self-attention in a way which appears to not be challenging. We also make general improvements to NVIB through changing the sampling to implicit reparameterisation gradients [(Figurnov et al. 2018)](https://proceedings.neurips.cc/paper_files/paper/2018/file/92c8c96e4c37100777c7190b76d28233-Paper.pdf)
> .  That said, we agree that, due to space constraints, we were not able to convey the range of alternatives which we had to consider when developing a successful application of NVIB to self-attention in stacked layers (see the next paragraph).  Instead we focused on presenting one model which was successful, which includes the layer-wise communication and a new loss structure to encourage the hierarchical abstraction. Finding this model was in fact challenging, which we will try to convey in future versions of the paper.
>
> >*Experiment: a more detailed explanation or exploration of why only these layers were chosen and how different configurations might impact the results.*
>
> **Hypothesis** Our experiments are designed to demonstrate the existence of a model which proves our hypothesis - **Applying NVIB to self-attention in stacked layers of a Transformer encoder leads to greater abstraction in the encoder.**   We did not include an ablation study of this proposed model simply because doing so would have required making this a long paper, and we believe that a better scientific strategy is to first demonstrate that this kind of architecture is effective and then study its properties in more detail in future work.
>
> **Design** That said, we would be happy to add more motivations for our design decisions in future versions of the paper.
> Following from [Henderson and Fehr (2023)](https://openreview.net/pdf?id=6QkjC_cs03X) and [Lewis et al. (2020)](https://aclanthology.org/2020.acl-main.703.pdf)
>  we designed a denoising reconstruction objective with a Transformer encoder-decoder structure. In *Appendix A* we define all the model details and that the encoders 6 layers are inspired by the base model of [Vaswani et al. (2017)](https://proceedings.neurips.cc/paper_files/paper/2017/file/3f5ee243547dee91fbd053c1c4a845aa-Paper.pdf). For the Transformer decoder we make the design choice to only use 2 layers such that the decoder is not able to overpower and ignore the encoders embeddings from cross attention.
>
> **Configuration** For the NVIB layers during experimentation we considered: All layer including NVIB; the last 3 layers including NVIB; and only the final layer including NVIB. When all layers were included it was challenging to get both compression and performance as the regularisation was too strong. Only regularising the last layer managed to reduce the number of vectors but often converged to a single sentence vector with lower, non-comparable validation cross-entropy. Finally, we settled on only regularising the last 3 layers as it gave the model enough flexibility in the lower layers and progressive compression in the higher layers. We do not think this was an exhaustive evaluation, but found a minimal configuration that proved our hypothesis.
>
> >  *Baseline: more advanced models*
>
> This paper is not claiming absolute improvements on the various evaluation measures, but instead is investigating an unsupervised method for learning abstract representations.  As such, the predefined levels of abstraction used in current advanced models, such as subword models and stride-based downsampling, are not relevant points of comparison. We instead report a controlled experiment to compare the effect of introducing NVIB on Transformer Encoders to Transformer Encoders without NVIB. To the best of our knowledge, a plain Transformer Encoder (the null abstraction) is the only baseline that is relevant to learned abstractions.
>
> **Preliminary Additional Experiment** That said, for the specific case of ArXiv topic classification, reporting results for a subword baseline would provide relevant context.  We conducted a preliminary experiment with BPE-based model. In particular, we trained a standard Trasformer autoencoder baseline with BPEs as inputs. We then performed the same experiment as described in section 4.3.1. Provisionally, we found that a BPE-based model achieved only 25.13% average F1, much worse than the reported results. We suspect this is due to the nature of words in the task is different to that of the tokenizer, which is exactly where our model shows its advantages. These results are provisional and would require more investigation to ensure precise comparability.
>
> > *Questions: For example, the right part in Figure 1 is not easy to understand. Does it mean different attention patterns at three different layers? In Figure 2, which color is “lighter” is a challenging task to tell.*
>
> Thank you! Your specific and constructive suggestions on Figure 1 and Figure 2 will be addressed.  In Figure 1 we have an illustrative example of how we would expect the attention maps of our NVIB-integrated encoder layers to appear. We shall update Figure 1 by finding an alternative to the attention maps and include in the Figure 2 caption that dark purple is 0 and light Yellow is 1 for the attention maps.

---

### Official Review · Reviewer_gprF · 2023-08-11

**Soundness:** 4

**Excitement:**

4: Strong: This paper deepens the understanding of some phenomenon or lowers the barriers to an existing research direction.

**Paper Topic And Main Contributions:**

This paper extends previous work by Henderson and Fehr (2023), which introduced the non-parametric informational bottleneck (NVBI) regularizer, to self-attention layers in generic transformers. This method consists of replacing the traditional self-attention mechanism with a Dirichlet process mixture model, where every vector is associated with a Gaussian mixture component, and whose mixture weights are drawn from a Dirichlet distribution. Crucially, the regularizer is setup to incentivize the sparsity of the mixture weight distribution. They train traditional and NVIB-regularized character-level transformers on Wikitext-2, and find that the attention weights at higher layers of the NVIB-regularized model better align with linguistically meaningful units of text (e.g., word), and in probing experiments they find they are better at capturing topic (on Arxiv topic classification task) and linguistic information (on SemEval), as well as being more robust to perturbations of the input sequence.

**Questions For The Authors:**

Question A: Is there a reason why you did not run experiments on language modeling? I understand that the current experiments control for some models to be trained better than others, but it would be interesting if this method also gestures towards better language models as a whole.

**Reasons To Accept:**

The paper is very well written. Their idea and contribution is explained clearly, and the NVIB regularizer, alongside the changes made to adapt it to stacked self-attention, are well described.

The idea is interesting, and the results are encouraging. The fact that the attention maps in higher layers better align with words, for example, is a very interesting finding, and the probing results suggest NVIB could lead to better representation learning as a whole.

**Reasons To Reject:**

It would perhaps be interesting to run some more experiments, in particular to see if NVIB regularization improves language modeling.

**Reproducibility:**

5: Could easily reproduce the results.

**Reviewer Confidence:**

3: Pretty sure, but there's a chance I missed something. Although I have a good feel for this area in general, I did not carefully check the paper's details, e.g., the math, experimental design, or novelty.

---

> ### Author Rebuttal · Authors · 2023-08-28
>
> We would like to thank Reviewer gprF for their time taken to understand our contributions and their encouraging remarks. We are honoured that they found our presentation: **very well written**, **clearly explained**  and **well described**. Our Results: **encouraging** with a **very interesting finding**.  Our idea: **interesting** and could lead to **better representation learning as a whole**. We address the reviewer's concern below.
>
> > *It would perhaps be interesting to run some more experiments, in particular to see if NVIB regularization improves language modeling.*
>
> We agree it would be interesting to run experiments on language modeling, which we are in fact currently working on. However, our experience is that language modeling requires a rather different architecture from the encoder-decoder architecture we use here (such as the requirement of a causal mask).  It would not be possible to add such experiments to this short paper.  In addition, we believe our existing experiments more directly support our claims. We appreciate the encouraging suggestion and will explore this in future work.

---

### Official Review · Reviewer_LT26 · 2023-08-11

**Soundness:** 3

**Excitement:**

3: Ambivalent: It has merits (e.g., it reports state-of-the-art results, the idea is nice), but there are key weaknesses (e.g., it describes incremental work), and it can significantly benefit from another round of revision. However, I won't object to accepting it if my co-reviewers champion it.

**Paper Topic And Main Contributions:**

This paper proposes to learn text representations at multiple levels of abstraction by applying a sparsity-inducing attention regularizer (NVIB) to the self-attention layers of a character-input transformer encoder.

List of contributions: NLP engineering experiment, reproduction study, publicly available software (upon publication).


**Questions For The Authors:**

It’s interesting how NVIB often organized latent representations around words/subwords in your experiments. I wonder if you discovered any larger constructions than words. Does the size of the discovered units continue to become larger as you increase the regularization or does it plateau after a certain point?

**Reasons To Accept:**

1) The authors propose adaptations of NVIB to self-attention and to use it for learning increasingly abstract representations at higher layers of a transformer encoder. (NVIB was originally introduced for cross-attention.)
2) The authors presented interesting results from intrinsic evaluation of the attention distributions of such a model pretrained on Wikitext-2 and from evaluating the quality of the representation in linguistic probing and text classification.
3) The authors compared the robustness of the NVIB-regularized model with that of a regular transformer and found the former more robust to noisy input.


**Reasons To Reject:**

1) The specific design choices made in adapting the original NVIB to self-attention could benefit from more justification and discussion.
e.g. Does it matter that the skip connection is multiplicative rather than additive? Does it matter to choose a strictly increasing strength of regularization as layers get higher? These could possibly be answered with ablation studies.


**Reproducibility:**

5: Could easily reproduce the results.

**Reviewer Confidence:**

2: Willing to defend my evaluation, but it is fairly likely that I missed some details, didn't understand some central points, or can't be sure about the novelty of the work.

**Typos Grammar Style And Presentation Improvements:**

It may help readability to give a little focus to the description of the big picture with NVIB in section 2.1.

---

> ### Author Rebuttal · Authors · 2023-08-28
>
> We really appreciate the time and effort from Reviewer LT26 to review our mathematics and model design. We would like to thank the reviewer for the kind remarks by highlighting our **interesting results** and our outlining our contributions in the reasons to accept. We address the reviewer's concerns and questions below.
>
> > *More justification and discussion of specific design choices*
>
> We are happy that the reviewer is so interested in the properties of our proposed model, but would like to emphasise that our first priority for this short paper is to demonstrate the existence of a model which proves our hypothesis - **Applying NVIB to self-attention in stacked layers of a Transformer encoder leads to greater abstraction in the encoder.**   Conducting a more thorough ablation study of the range of model design choices would have required a long paper.  When our first result has been accepted, then we will be in a position to explore the range of effective models in future work.
>
> That said, to the extent that space permits, we would be happy to add more justification and discussion of specific design choices in future versions of this paper.  In particular, based on our experience with exploring the space of models so far, the very interesting specific questions raised by the reviewer can be addressed, as follows.
>
> > *Does it matter that the skip connection is multiplicative rather than additive?*
>
> Yes. The multiplicative skip connection of NVIB Dirichlet pseudo-count parameters $\boldsymbol{\alpha}^{(l)}$ from the previous layer $(l-1)$ was included for two reasons:
> 1. **Overflow** The pseudo-count parameters $\boldsymbol{\alpha}$ control the noise in sampling during training and bias the attention weights. During initial training steps with KL annealing the model is incentivised to make the $\boldsymbol{\alpha}$ larger. This is due to low initial regularisation in KL annealing. For overflow and precision the original NVIB implementation kept the $\boldsymbol{\alpha}$ in log space, thus we thought it would be better to keep the skip connection in log space.
> 2. **Magnitude** The skip connection was put in place to allow communication in attention between layers and guide the abstraction. Since we considered only 3 layers of NVIB a multiplicative skip connection has a greater magnitude than an additive skip connection. Considering an additive skip connection is a valuable suggestion and we suspect that this may be helpful when considering many more connected NVIB layers.
>
> > *Does it matter to choose a strictly increasing strength of regularization as layers get higher?*
>
> Yes. During initial exploration of the regularisation we found a uniform regularisation strength across layers did not lead to the same abstraction. Often the lower levels were penalised too strongly. We also considered a regularisation that would double in strength through the layers, which is more in line with the multiplicative skip connection. This regularisation was too strong and lead to the final layer being compressed to a single sentence representation.
>
> > *Does the size of the discovered units continue to become larger as you increase the regularization or does it plateau after a certain point?*
>
> Yes. The size of the representations become larger until all units are compressed into a single vector, much like a sentence embedding. This single vector made it challenging to get competitive reconstruction cross-entropy during validation. If you over-regularise, the representations collapse to the uniformed prior representation from NVIB (as is typical of VIB models). The model will then try to solve the task with the decoder alone.
>
> > *It may help readability to give a little focus to the description of the big picture with NVIB in section 2.1.*
>
> We agree that writing Section 2.1 within the space constraints was very challenging, and welcome suggestions for improvement.  We intend to follow the advice from other reviewers to add more detail to this section, which is currently in the Appendix C and D.  We agree that keeping in mind a focus on the relevant properties of NVIB will be important when expanding this section.  If Reviewer LT26 has a different interpretation of "focus" in mind, we would welcome more comments on this.

---

### Official Review · Reviewer_YN7E · 2023-08-13

**Soundness:** 3

**Excitement:**

2: Mediocre: This paper makes marginal contributions (vs non-contemporaneous work), so I would rather not see it in the conference.

**Paper Topic And Main Contributions:**

This paper is about introducing a novel language representation model called Nonparametric Variational Information Bottleneck (NVIB) that can learn to compress to different levels of abstraction at different layers of the same model. The paper addresses the problem of the high cost of learning textual embeddings, which are tokenization specific and require different models to be trained for each level of abstraction. The main contribution of this paper is the introduction of NVIB, which can learn to compress to different levels of abstraction at different layers of the same model, resulting in a more linguistically informed and robust model. The paper also presents experimental results that demonstrate the effectiveness of NVIB in various NLP tasks, including classification and linguistic probing.

**Questions For The Authors:**

1. How is the decoding and training efficiency of this model compared with sub-word based language model? Especially for current LLM era where sequence length is extremely important.
2. How is the expected performance of this method on other languages such as Japanese and Chinese which has a much bigger character size?

**Reasons To Accept:**

The strengths of this paper include the introduction of a novel regularization term for character based language representation model, Nonparametric Variational Information Bottleneck (NVIB), that can learn to compress to different levels of abstraction at different layers of the same model, resulting in a more linguistically informed and robust model. The paper also presents experimental results that demonstrate the effectiveness of NVIB in various NLP tasks, including classification and linguistic probing.

**Reasons To Reject:**

The paper has some weaknesses. Firstly, the experimental section is relatively simple, and the authors could benefit from comparing their method with more baselines, including sub-word based methods, to demonstrate the benefits of their approach more convincingly. This would provide a more comprehensive evaluation of the proposed model and its effectiveness in comparison to other state-of-the-art methods.

Secondly, the paper does not provide sufficient information on the decoding and training efficiency of the proposed model compared to sub-word based language models. This is particularly important in the current era of large language models where sequence length is a crucial factor. A more detailed analysis of the computational efficiency of the proposed model would be useful to understand its practicality and scalability in real-world applications.

Finally, the paper only evaluates the proposed model on English text, and it would be interesting to see how it performs on other languages such as Japanese and Chinese, which have a much larger character size. This would provide insights into the generalizability of the proposed model and its effectiveness in capturing the linguistic properties of different languages.

**Reproducibility:**

4: Could mostly reproduce the results, but there may be some variation because of sample variance or minor variations in their interpretation of the protocol or method.

**Reviewer Confidence:**

3: Pretty sure, but there's a chance I missed something. Although I have a good feel for this area in general, I did not carefully check the paper's details, e.g., the math, experimental design, or novelty.

---

> ### Author Rebuttal · Authors · 2023-08-28
>
> We thank Reviewer YN7E for taking the time to review our short paper, but it appears that the reviewer does not have an accurate perception of the nature and scope of our claimed contributions. As explained in the **Contributions** paragraph on page 1, we claim to have contributed:
> 1. A novel application of NVIB [(Henderson and Fehr, 2023)](https://openreview.net/pdf?id=6QkjC_cs03X) to stacked self-attention layers of a Transformer encoder.
> 2. Applying NVIB to self-attention in stacked layers of a Transformer encoder leads to greater abstraction in the encoder - Shown  qualitatively with attention plots and quantitatively in F1 segment overlaps.
> 3. Semantically meaningful representations - Shown with both ArXiv Topic Classification and SentEval probing tasks.
> 4. Robustness - Shown with input perturbation plots.
>
> In light of this, we would like to clarify the following points:
> 1. We did not introduce NVIB; we only applied it to stacked self-attention layers of a Transformer encoder.
> 2. NVIB is not a regularisation term but a information theoretic regularisation for the entire latent representation of Transformers.
> 3. We are not trying to show practical effectiveness on NLP tasks, but to test our scientific hypotheses.
>
> According to the [ACL reviewing guidelines](https://2023.aclweb.org/blog/review-form/), Soundness is defined by *"Does the paper clearly state scientific claims and provide adequate support for them?"*.  Since Reviewer YN7E does not provide any judgement against our specific claims, we feel the **Soundness score of 1 is unjust.**  Proposing different claims that the reviewer thinks we should have made and then arguing that we did not show them is not relevant for Soundness.
>
> We shall now address the weaknesses and concerns of the Reviewer:
>
> > *The experimental section is relatively simple.*
>
> Yes, the simple experimental design is important and a strength. The experiments are designed to find a minimal model to prove our hypothesis - **Applying NVIB to self-attention in stacked layers of a Transformer encoder leads to greater abstraction in the encoder**. We conducted a controlled experiment to compare the effect of introducing NVIB on Transformer Encoders to Transformer Encoders without NVIB.
>
> > *Comparing their method with more baselines, including sub-word based methods.*
>
> This paper is not claiming absolute improvements on the various evaluation measures, but instead is investigating an unsupervised method for learning abstract representations.  As such, predefined levels of abstraction such as subword models are not relevant points of comparison. We instead report a controlled experiment to compare the effect of introducing NVIB on Transformer Encoders to Transformer Encoders without NVIB. To the best of our knowledge, a plain Transformer Encoder (the null abstraction) is the only baseline that is relevant to learned abstractions.
>
> **Preliminary Additional Experiment** That said, for the specific case of ArXiv topic classification, reporting results for a subword baseline would provide relevant context.  We conducted a preliminary experiment with BPE-based model. In particular, we trained a standard Trasformer autoencoder baseline with BPEs as inputs. We then performed the same experiment as described in section 4.3.1. Provisionally, we found that a BPE-based model achieved only 25.13% average F1, much worse than the reported results. We suspect this is due to the nature of words in the task is different to that of the tokenizer, which is exactly where our model shows its advantages. These results are provisional and would require more investigation to ensure precise comparability.
>
> While it would be possible to use subword models as a base model on which we apply NVIB, it is not clear what level of abstraction we should expect to find.  In contrast, it is relatively clear that the next level of abstraction above the character level is something like morphemes or words.  For this reason, we conducted all our experiments starting from character-level inputs.
>
> > *The paper does not provide sufficient information on the decoding and training efficiency.*
>
> We agree that a detailed analysis of the computational efficiency would be useful.  In theory, there should be a substantial improvement in the $O(N^2)$ complexity of attention because NVIB substantially reduces an $N$ (by dropping key vectors) at higher layers of the Transformer.  However, given the scope of this short paper, the fact that we submitted to the interpretability track and our claims, we feel this is not a weakness.
>
> **Additional Analysis** For completeness we ran extra experiments with our current implementation to access the computational costs overhead of NVIB. Including the NVIB layer adds 33\% more parameters per attention layer and only adds **10\% more parameters** to our whole model. We compared the computational cost of including NVIB layer and attention over standard attention. We used input lengths of 8, 512 and 4096 and found **less than 1\% increase in computational costs** during: training and inference; across all lengths; and across all metrics (memory, FLOPS and GPU time). Hence the computation is still dominated by $O(N^2)$, for our current implementation. Considering the full model, over a single epoch, NVIB is actually **20\% faster during training** and **15\% faster during autoregressive decoding** than the full Transformer baseline. However, due to our KL annealing training schedule and regularisation we found that training with NVIB required more steps to reach a similar validation cross-entropy (Appendix A, Table 2).
>
> > *Only evaluates the proposed model on English text.*
>
> We agree that it would be interesting to check the effectiveness in capturing the linguistic properties of different languages. Our limitations state *"In addition, the experiments are only done on English, but we would expect more improvements with more morphologically rich languages"*.  But additional languages are not required to demonstrate the existence of NVIB models which support our claims.  As stated in the [ACL reviewing guidelines](https://2023.aclweb.org/blog/review-acl23/#1-acl-2023-review-form): *"Any other extra experiments are in the "nice-to-have" category, and belong in the "suggestions" section rather than "reasons to reject." This heuristic is particularly damaging for short papers."*  Also, according to [Reviewing for EMNLP](https://2020.emnlp.org/blog/2020-05-17-write-good-reviews), this is an invalid bases for rejecting a paper: *"We care about NLP for any language."*
>
> > *How is the expected performance of this method on other languages such as Japanese and Chinese which has a much bigger character size?*
>
> The fact that these characters (for Japanese and Chinese) represent larger chunks of language makes them less interesting for our experiments for the same reason given above for not using subword inputs, namely that it is less clear what the larger abstract units should be. Also, these languages are not morphologically rich and vocabulary size would not affect the proposed attention regularisation mechanism. They would be interesting languages for any future work which addresses the usefulness of NVIB at higher levels of abstraction or for practical downstream tasks, for example for finding sentence groupings for summarisation or question answering.

---

### Meta-Review · Area_Chair_8LWc · 2023-09-19

**Recommendation:** 4

**Metareview:**

There is consensus amongst the reviewers after the rebuttal period that the presented work is Sound and moderately exciting. It is clear that the methodology itself is exploring novel perspectives, and can shed further light on the internal workings by understand how abstraction of knowledge is represented in a model. Most of the weaknesses point towards a limited set of experimentation and exploration. However, being a short paper, the experiments necessarily have to be focused and directed towards a concrete hypothesis, which the work successful does. Overall, the paper has _good technical soundness_ and _moderate excitement_.

The following is a summary of the strengths, weaknesses and scores across the six reviews:

**Strengths:**

- Proposed methodology (NVIB applied to transformers) is more linguistically motivated towards robustness (**YN7E**, **LT26**, **gprF**, **2hkB**, **fPkp**, **bGq2**)
- Experimental results back the efficacy of the proposed method well across several NLP tasks (**YN7E**, **LT26**, **2hkB**, **bGq2**)
- The main results and overall contributions are well presented (**gprF**, **2hkB**)

**Weaknesses:**

- Design choices and technical contributions are not well motivated, and sufficient prior-knowledge is not communicated well (**LT26**, **bGq2**)
	- Authors agreed to add more details in the rebuttal, and alluded to the lack of space in a short paper for this.
- Experimental setup is limited (e.g. characters only, single language, small alphabet size, specific task-tuned models) (**YN7E**, **gprF**, **fPkp**)

**Scores in decreasing order of confidence:**

|      | Soundness | Excitement | Reproducibility | Confidence |
|------|-----------|------------|-----------------|------------|
| YN7E | 3         | 2          | 4               | 3          |
| gprF | 4         | 4          | 5               | 3          |
| fPkp | 3         | 2          | 3               | 3          |
| LT26 | 3         | 3          | 5               | 2          |
| 2hkB | 3         | 3          | 4               | 2          |
| bGq2 | 3         | 3          | 3               | 1          |

---

### Decision · Program_Chairs · 2023-10-07

**Decision:**

Accept-Findings

**Comment:**

There is consensus amongst the reviewers after the rebuttal period that the presented work is Sound and moderately exciting. It is clear that the methodology itself is exploring novel perspectives, and can shed further light on the internal workings by understand how abstraction of knowledge is represented in a model. Most of the weaknesses point towards a limited set of experimentation and exploration. However, being a short paper, the experiments necessarily have to be focused and directed towards a concrete hypothesis, which the work successful does. Overall, the paper has _good technical soundness_ and _moderate excitement_.

The following is a summary of the strengths, weaknesses and scores across the six reviews:

**Strengths:**

- Proposed methodology (NVIB applied to transformers) is more linguistically motivated towards robustness (**YN7E**, **LT26**, **gprF**, **2hkB**, **fPkp**, **bGq2**)
- Experimental results back the efficacy of the proposed method well across several NLP tasks (**YN7E**, **LT26**, **2hkB**, **bGq2**)
- The main results and overall contributions are well presented (**gprF**, **2hkB**)

**Weaknesses:**

- Design choices and technical contributions are not well motivated, and sufficient prior-knowledge is not communicated well (**LT26**, **bGq2**)
	- Authors agreed to add more details in the rebuttal, and alluded to the lack of space in a short paper for this.
- Experimental setup is limited (e.g. characters only, single language, small alphabet size, specific task-tuned models) (**YN7E**, **gprF**, **fPkp**)

**Scores in decreasing order of confidence:**

|      | Soundness | Excitement | Reproducibility | Confidence |
|------|-----------|------------|-----------------|------------|
| YN7E | 3         | 2          | 4               | 3          |
| gprF | 4         | 4          | 5               | 3          |
| fPkp | 3         | 2          | 3               | 3          |
| LT26 | 3         | 3          | 5               | 2          |
| 2hkB | 3         | 3          | 4               | 2          |
| bGq2 | 3         | 3          | 3               | 1          |